# Controlled Delivery of BET-PROTACs: In Vitro Evaluation of MZ1-Loaded Polymeric Antibody Conjugated Nanoparticles in Breast Cancer

**DOI:** 10.3390/pharmaceutics12100986

**Published:** 2020-10-19

**Authors:** Francisco J. Cimas, Enrique Niza, Alberto Juan, María del Mar Noblejas-López, Iván Bravo, Agustín Lara-Sanchez, Carlos Alonso-Moreno, Alberto Ocaña

**Affiliations:** 1Oncología Traslacional, Centro Regional de Investigaciones Biomédicas, 02008 Albacete, Spain; FranciscoJose.Cimas@uclm.es (F.J.C.); MariadelMar.Noblejas@uclm.es (M.d.M.N.-L.); 2Oncología Traslacional, Unidad de Investigación del Complejo Hospitalario Universitario de Albacete, 02008 Albacete, Spain; alberto.juan@uclm.es; 3Centro Regional de Investigaciones Biomédicas, Unidad NanoCRIB, 02008 Albacete, Spain; enrique.niza@uclm.es (E.N.); ivan.bravo@uclm.es (I.B.); 4Facultad de Farmacia de Albacete, Universidad de Castilla-La Mancha, 02008 Albacete, Spain; 5Facultad de Ciencias y Tecnologías Químicas, Universidad de Castilla-La Mancha, 13005 Ciudad Real, Spain; Agustin.lara@uclm.es; 6Experimental Therapeutics Unit, Hospital clínico San Carlos, IdISSC and CIBERONC, 28029 Madrid, Spain

**Keywords:** HER2, breast cancer, trastuzumab, nanoparticles, JQ1-based PROTAC, MZ1

## Abstract

Bromo and extraterminal domain (BET) inhibitors-PROteolysis TArgeting Chimera (BETi-PROTAC) is a new family of compounds that induce proteasomal degradation through the ubiquitination of the tagged to BET inhibitors Bromodomain proteins, BRD2 and BRD. The encapsulation and controlled release of BET-PROTACs through their vectorization with antibodies, like trastuzumab, could facilitate their pharmacokinetic and efficacy profile. Antibody conjugated nanoparticles (ACNPs) using PROTACs have not been designed and evaluated. In this pioneer approach, the commercial MZ1 PROTAC was encapsulated into the FDA-approved polymeric nanoparticles. The nanoparticles were conjugated with trastuzumab to guide the delivery of MZ1 to breast tumoral cells that overexpress HER2. These ACNPs were characterized by means of size, polydispersity index, and Z-potential. Morphology of the nanoparticles, along with stability and release studies, completed the characterization. MZ1-loaded ACNPs showed a significant cytotoxic effect maintaining its mechanism of action and improving its therapeutic properties.

## 1. Introduction

Antibodies and small tyrosine kinase inhibitors (TKIs) have been largely used for cancer treatment [1]. Antibodies target proteins located outside the cellular membrane, thus they act mostly on proteins that are present in the tumoral cell membrane or in the interstitial space, which limits their effect on oncoproteins that act in the intracellular or nuclear compartment [2,3]. On the other hand, TKIs have a favorable permeability acting on proteins with enzymatic activity that are located within these compartments. However, the major disadvantage of TKIs for clinic purposes relies on the easy generation of drug resistance after a long-term administration [4]. PROteolysis TArgeting Chimera (PROTAC) have raised high expectations in the field as could overcome such limitations [5]. PROTACs with a warhead able to bind the mutated kinase could result in a stable interaction, potentially rescuing the classical mechanism of resistance to kinase inhibitors [6]. Resistance due to mutations in proteins, which lack enzymatic activity, can also be bypassed by PROTACs. Thus, PROTACs targeting the ER in breast cancer could rescue resistance to anti-estrogens when this resistance is mediated by mutations at the ER.

PROTACs need to be configurated by a E3 ligase-recruiting ligand, a POI-binding ligand, and a linker [7,8]. Thus, PROTACs mediate the transferring of ubiquitin onto the POI, resulting in degradation [9]. In this context, PROTACs can act on several proteins with different biological roles, as long as there is a POI-binding ligand for that protein. ARV-110 and ARV-471 are PROTACs in phase I clinical trials (NCT03888612 and NCT04072952) for prostate and breast cancer, respectively [10,11]. ARV-110 degrades androgen receptor (AR) as well as multiple clinically relevant AR mutants, whereas ARV-471 targets ERα and clinically relevant ERα mutants (Y537S and D538G). ARV-471 is being evaluated in clinical trials for women with metastatic ER+ positive/HER2− negative breast cancer [12].

Some PROTACs have been reported in the literature in relation to breast cancer therapy. ERD-308 is an ER degrader, which, compared to fulvestrant, showed a stronger inhibitory effect on proliferation in MCF-7 cells [13]. Small-molecule pan-BET degraders such as ARV-771 and dBET1 were designed to induce superior effects in breast cancer [14,15]. BETd-246 was reported to degrade BET proteins in triple-negative breast cancer (TNBC) [16]. This PROTAC exhibited a strong inhibitory effect in several TNBC cell lines accompanied by time-dependent downregulation of MLC1. MZ1 is a selective BRD4 degrader over BRD2 and BRD3 proteins via the proteasome [17,18,19] and have shown high activity in some hematological malignancies [20]. Furthermore, we have previously reported the activity of this molecule in a breast cancer model of TNBC [21].

The designing of PROTACs has different limitations, including problems in isolation, reduced cell permeability, and off-target toxicity effects due to a low therapeutic index [22,23,24]. PROTACs act in a catalytic mode, which hampers their pharmacokinetic and pharmacodynamic evaluation. In this context, studies concerning degradation activity, selectivity, and toxicity are still required for the optimal design of PROTACs. Clearly, novel strategies in this field must be aggressively pursued for a faster clinical translation.

Nanomedicine may have the potential to improve novel therapeutics for the treatment of solid tumors by optimizing the release of PROTACs through the use of vectorized nanocarriers. Very recently, Wang et al. designed a novel nanoparticle-based PROTAC platform for the ALK target degradation as a potential therapeutic strategy to treat EML4-ALK-positive diseases [25]. This article was the proof of concept for the incorporation of PROTACs into nanoparticles. A further next step is the development of ACNPs as a combination of the potential of antibody conjugation and nanotechnology [26,27]. Antibodies against membrane proteins expressed in tumoral cells can be used as vectors. Such antibodies can be conjugated to the NPs to deliver drugs in a controlled manner, preserving its chemical structure, avoiding unpredicted metabolization, and reducing toxicity. The use of ACNPs might improve tumor accumulation, target selectivity, and cytotoxic efficiency for the treatment of breast cancer.

After reporting on the high activities of MZ1 in BETi sensitive and resistant breast cancer cells, evaluating its mechanism of action, and the in vivo activity [21], here we aim to explore the delivery of MZ1 for breast cancer therapy. For this purpose, MZ1-loaded ACNPs conjugated with trastuzumab were developed, characterized, and evaluated in vitro against a panel of different breast cancer cells.

## 2. Materials and Methods

### 2.1. Materials

Poly-d-l-Lactice (22,000 Da) (PLA) was synthesized under nitrogen atmosphere, using standard Schlenk techniques [28] (see NMR characterization in Appendix A). ZnEt_2_ 1M solution (Sigma-Aldrich, Madrid, Spain) was used as received. L-LA (Sigma-Aldrich) was sublimed 3 times and stored in the glovebox. Zinc catalyst was prepared according to literature procedures [29]. (+)-JQ1 based PROTAC (MZ1) were purchased (high-performance liquid chromatography (HPLC) ≥98% purity) by TOCRIS Bioscience (Bristol, UK), and trastuzumab was purchased as Herceptin by ROCHE (F. Hoffmann-La Roche Ltd.). N-(3-Dimethylaminopropyl)-N′-ethylcarbodiimide hydrochloride (EDC) ((HPLC) ≥ 98% purity) and N-Hydroxysuccinimide (NHS) ((HPLC) ≥98% purity) were purchased from sigma Aldrich (Madrid, Spain).

### 2.2. Characterization

Proton nuclear magnetic resonance (Varian Inova FT-500 spectrometer) and gel permeation chromatography (PL-GPC-220 instrument) were used for polymer characterization. NMR spectra were recorded on a Varian Inova FT-500 spectrometer. The morphology of the nanoparticles was characterized by using Transmission Electron Microscopy (TEM). Zetasizer Nano ZS (Malvern Instruments) and Jeol JEM 2100 transmission electronic microscope (Oxford Link EDS detector and Digital MicrographTM software from Gatan) were employed for nanoparticle characterization.

The efficiency of the Loading (LE) and encapsulation (EE) were calculated according to:LE % = (weight of encapsulated MZ1 (mg))/(weight of total (MZ1 encapsulated + scaffold weight) (mg)) × 100%(1)
EE % = (weight of encapsulated MZ1 (mg))/(weight of MZ1 feeding (mg)) × 100%(2)

### 2.3. Preparation of Nanoparticles (NPs)

MZ1-loaded NPs (MZ1-NPs). The Nanoparticles (NPs) were prepared by nanoprecipitation and the displacement solvent method. Briefly, 10 mg of PLA in 1 mL of acetone and 5 mg of MZ1 in 70 μL of DMSO was mixed and sonicated for 10 min to form the organic phase. The organic phase and 17 mL of an aqueous phase (0.5% PEI and 1%PVA) were mixed up under slow stirring. The acetone from the organic phase was evaporated under reduced pressure and centrifuged at 15,000 rpm for 20 min at 4 °C.

MZ1-loaded ACNPs (MZ1-ACNPs). Trastuzumab was chemically conjugated to PEI coating NPs via carbodiimide chemistry [30]. Briefly, tratuzumab (21 mg mL^−1^ in 0.1 M PBS, pH 7.4) were activated in 4 mL of PBS (0.1 M, pH 5.8) using 40 mg of 1-ethyl-3-(3-dimethylaminopropyl) carbodiimide (EDC) and 9.7 mg of N-hydroxysuccinimide (NHS). Then, PEI coating NPs suspension in PBS pH 5.8 was added to the activated trastuzumab and left for 12 h (room temperature). Finally, MZ1-ACNPs were collected after centrifugation 15,000 rpm for 20 min (4 °C).

Conjugation quantification. Standard protocol of the Bradford assay (BCA) was employed for quantifying the concentration of antibodies. Briefly, ACNPs samples were incubated in a 96 well plate with a BCA solution for 30 min in the dark. Then, the supernatant was taken out to measure the concentration of non-conjugated antibodies using a spectrophotometer at 563 nm.

Stability of NPs. The NPs were incubated at 37 °C (1 mg·mL^−1^) in PBS, and the average size (nm) and polydispersity index (PdI) determined over time by dynamic light scattering (DLS) measurements.

Drug-release studies. Lyophilized nanoparticles were sealed in a dialysis membrane (3500 Da) and suspended in 10 mL of phosphate-buffered saline (pH 7.4 and pH 5.8). 3 mL of release medium was taken out and replaced by the fresh medium at certain intervals to measure the concentration using a spectrophotometer at 275 nm. The experiment was carried out in 3 replicates.

Protein corona formation. MZ1-NPs and MZ1-ACNPs were incubated with bovine serum albumin (BSA) (1:1 in weight) for 35 min at 37 °C. As a control, the equivalent volume of Phosphate-buffered saline (PBS) was added to the same amount of BSA. Then, samples were centrifuged at 14,000 rpm for 20 min at 4 °C in order to precipitate the nanoparticles. Finally, the supernatant was collected, and the protein concentration was measured by the Pierce BCA assay kit (ThermoFisher) following the manufacturer’s instructions.

### 2.4. In Vitro Assays

Cell culture. The HER2+ cell lines (BT474, SKBR3 and HCC1954) were grown using Dulbecco’s modified Eagle medium (DMEM) with 10% inactivated fetal bovine serum, 2 mM L-glutamine, penicillin (20 units mL^−1^) and streptomycin (5 μg mL^−1^) and maintained at 37 °C in a saturated humidity atmosphere with 5% of CO_2_. These cell lines were kindly provided by Drs. J. Losada and A. Balmain, which were acquired from ATCC in 2015. The cell lines were genetically confirmed by STR (Salamanca University Hospital).

Viability. Cells were seeded onto 48-well plates at a dilution of 10,000 cells by well, for 24 h in the conditions described before [30]. HER2+ cells were treated with vehicle (Ctl), MZ1 (50, 100, 200 and 400 nM), MZ1- NPs (5, 25, 50 and 100 nM) and and MZ1-ACNPs (5, 25, 50 and 100 nM of MZ1). After 72 h, 3-(4,5-dimethylthiazol-2-yl)-2,5-diphenyl tetrazolium bromide (MTT) (5 mg/mL) was added into each well and were incubated for 60 min at 37 °C. Following, the culture medium was removed, and the formed formazan crystals dissolved in 200 μL DMSO (Merck Millipore, Spain). To evaluate changes in viability, the absorbance of the dilution was measured at 555 nm using a multiwell plate reader (BMG labtech, Ortenberg, Germany).

Cell-cycle studies. Cells were planted in 6-well plates up to 250,000 cells by well. After 24 h, cells were treated with vehicle (Ctl), MZ1, MZ1- NPs, and MZ1-ACNPs (50 nM of MZ1) for 48 h. Cells were detached and fixed using 70% cold ethanol for 30 min at 4 °C. Next, cells were rinsed with PBS + 2% BSA and stained with propidium iodide staining solution (Immunostep S.L.). Staining was measured and analyzed in FACSCanto II flow cytometer unit from BD Biosciences.

Apoptosis. Cells were planted in 6-well plates up to 250,000 cells by well. After 24 h, cells were treated with vehicle (Ctl), MZ1, MZ1- NPs, and MZ1-ACNPs (50 nM of MZ1) for 72 h. After that, cells were collected and stained at room temperature and protected from light using propidium iodide (2 mg mL^−1^) and Annexin V-DT-634 (Immunostep S.L.) for 1 h. Cell death was evaluated in a FACSCanto II flow cytometer (BD Biosciences). For representation, cells were divided in 2 populations, the living (Annexin and PI negatives) and dead cells (Annexin and/or PI positive).

Statistical analysis. Data were represented as mean ± s.e.m. from 3 independent experiments. Statistics analysis was develop using GraphPad Prism software (version 6.1). In order to identify statistically significant differences between treatments, the unpaired t-test for independent samples was used, considering a level of significance of 95%, using the following legend: * *p* ≤ 0.05; ** *p* ≤ 0.01 and *** *p* ≤ 0.001.

## 3. Results

### 3.1. MZ1-Loaded Trastuzumbab Conjugated NPs (MZ1-ACNPs)

Polylactide (PLA) and Polyethyleneimine (PEI), FDA-approved polymers, were selected as building blocks for the generation of MZ1-loaded NPs (MZ1-NPs), as described elsewhere [30]. NPs were loaded with MZ1 and conjugated with Trastuzumab by covalent coupling via zero-cross-linker carbodiimide chemistry to generate MZ1-loaded ACNPs (Scheme 1).

Characterization of NPs and ACNPs was carried out by DLS and electronic microscopy (see Table 1, Figure 1, and Appendix A). DLS studies showed an average particle size for formulations close to 100 nm. The standard protocol of Bradford assay was employed for quantifying the concentration of the antibody in the supernatant (see Section 2). 1.6 nM was the trastuzumab cargo over the NP surface selected in accordance with previously published results [30]. The trastuzumab conjugation was confirmed by the decrease in the surface charge of NPs (Z-potential) from +46.3 mV (MZ1-NPs) to +31.8 mV (MZ1-ACNPs). TEM images showed nanoparticles of the very similar size reported by DLS measures and a core-shell morphology. After conjugation with trastuzumab, the surface of the NPs was modified (Figure 1).

Loading (%LE) and encapsulation efficiency (%EE) of MZ1-NPs and MZ1-ACNPs are collected in Table 1. MZ1-NPs showed higher %EE and LE% than MZ1-ACNPs formulations. It seems that the conjugation strategy, which required activation of the NPs prior to conjugation causes the release of the drug by diffusion.

The physical stability of MZ1-ACNPs was also studied in phosphate buffer saline (PBS). The values of average size (nm) and PdI of the MZ1-ACNPs were monitored by DLS (Figure 2B). The negligible increase in either particle size or PdI during a 7-day long experiment denoted high stability against aggregation.

In vitro release of both formulations was carried out using the dialysis method at pH 7.4 to mimic the physiological pH of circulation. Furthermore, the release profile for the MZ1-ACNPs at acid conditions was also carried out (see Appendix A). Figure 2A shows the biphasic profile for both loaded polymeric NPs at physiological pH [31]. A very short burst release and an 8 h diffusion step are reported for both formulations. As illustrated in Figure 2A, MZ1-ACNPs exhibited a controlled release of MZ1, slower than that observed for MZ1-NPs. The antibody conjugation over the surface of the nanoparticles might delay the diffusion of the drugs due to the most difficult accessibility of the water molecules to the core of the nanoparticles.

Experiments of protein corona analysis were carried out for MZ1-NPs and MZ1-ACNPs. Noteworthy, no formation of hard plasma protein corona was observed for MZ1-ACNPs (see Section 2 for methodology and Appendix A). The existence of antibodies over the surface of the ACNPs could prevent protein corona formation. This observation is of high importance because protein corona could affect the biodistribution and specificity of the delivery process.

### 3.2. Cytotoxic Effect

The cytotoxic effect of MZ1-ACNPs on tumoral cells was assayed by monitoring their ability to inhibit cell growth using MTT assays in two cell lines representative of HER2+ breast cancer with different sensitivity to MZ1, SKBR3 (more resistant), and BT474 (more sensitive). As we have previously shown, non-loaded NPs and ACNPs did not display any significant cytotoxicity in tumoral cells [30]. Therefore, cells were treated with vehicle, MZ1, MZ1-NPs, and MZ1-ACNPs for 72 h at different doses (Figure 3). Interestingly, MZ1-ACNPs had a significantly improved cytotoxic effect in comparison to free MZ1 and MZ1-NPs, both rendering similar results.

### 3.3. Cell Cycle Arrest and Apoptosis

Given the fact that MZ1-ACNPs inhibited cell proliferation in HER2+ breast cancer cell lines, we next explored their mechanism of action. Thus, SKBR3 and BT474 were treated with vehicle, MZ1, MZ1-NPs, and MZ1-ACNPs for 48 h and 72 h for cell cycle and induction of apoptosis analysis, respectively. Concerning the cell cycle, the ACNPs did not show any statistically significant difference in relation to the distribution of the cell cycle phases (Figure 4A,B). On the other hand, the results concerning the induction of cell death, with a dose-treatment of MZ1-ACNPs of 50 nM, indicated a remarkable increase in the induction of apoptosis compared with the non-vectorized MZ1-NPs vehicle (Figure 4C,D).

### 3.4. Cytotoxic Effect in HER2+, MZ1-Resistant Cell Lines

Due to the strong cytotoxic effect observed with MZ1-ACNPs in the most MZ1-resistant cell line, its effect in a naturally MZ1-resistant HER2+ cell line, HCC1054, was assessed. MZ1-ACNPs were able to bypass natural resistance to this PROTAC, being the NPs more cytotoxic for this cell line after conjugation with trastuzumab (Figure 5).

## 4. Discussion

Trastuzumab, pertuzumab, and T-DM1 are approved antibodies for the treatment of HER2+ breast cancer. Small molecule inhibitors of the kinase activity of the receptor, such as lapanitib or neratinib are also approved therapy for this disease. Both treatments improve clinical outcomes. However, there are still many patients who become resistant to treatment. Hence, HER2+ breast cancer remains cataloged as an incurable condition [32].

Identification of novel and druggable targets remains a top priority for the pharma/biotech industry. PROTACs have great potential for therapeutic intervention [33], and their mechanism is based on the inhibition of protein function by hijacking a ubiquitin E3 ligase for protein degradation. However, the lack of tumoral selectivity is expected to make PROTACs toxic, mainly if the degraded protein is promiscuously expressed [7,34]. PROTACs have shown the capability of degrading specific Bromodomains like BRD2 and BRD4 that play a key role in cancer progression. In breast cancer, the degradation of the protein has shown more efficacy than target inhibition [21]. The incorporation of PROTACs in ACNPs might offer the opportunity to overcome pharmacokinetic limitations and ensure their delivery at the desired site in the required proportions. The variety of drug agents that can be incorporated into ACNPs offers further development opportunities than can be afforded with other strategies such as standard antibody-drug conjugate technologies. In this context, developing ACNPs based on PROTAC-loaded for the treatment of breast cancer must be considered. As a proof of concept, MZ1 PROTAC was encapsulated in trastuzumab-conjugated NPs with the aim of improving pharmacokinetic issues.

As far as we know, nanotechnology has not been explored for PROTAC delivery. Only the recent work from Wang et al. provides evidence for the development of a new concept for PROTAC design [25]. They prepared a nanoplatform for ALK target degradation. Ceritinib was the ALK binding moiety and pomalidomise was used as E3ligase ligand. This nanoplatform displayed potent target degradation potency in NCI-H2228 cells in a dose -and time-dependent manner in vitro. In vivo data were not reported in this article.

In this work, the MZ1-ACNPs obtained were characterized by size, PdI, and superficial charge (Table 1 and Figure 1). The average ACNPs size was 114 nm, which is in the same range reported for other authors for drug encapsulation in polymeric nanoparticles [35]. Once MZ1-NPs formulation was optimized, reaching an EE value of 50%, trastuzumab conjugation was carried out following covalent binding by EDC/NHS chemistry. The ACNPs generation was carried out following previous works in a similar way to the previous work reported by our laboratory, where dasatinib was successfully encapsulated and evaluated for breast cancer therapy [30]. The low values for LE in comparison to dasatinib-loaded ACNPs counterparts could be justified by the higher hydrophilicity of MZ1. In contrast, smaller average size and Z-potential values were obtained in MZ1 encapsulation. The stability of MZ1-ACNPs were monitored by changes on R_H_ and PdI values. The results showed high stability over time, which preserves aggregation phenomena (Figure 2B). Negligible burst release and controlled release over time driven by diffusion were reported for these ACNPs (Figure 2A). Both the stability and controlled release are crucial to translate findings into the clinic.

On the other hand, the cytotoxicity studies indicate that the encapsulation of MZ1 in nanoparticles and ulterior conjugation with trastuzumab improved antitumoral effects in over-expressing HER2+ breast cancer cell lines, such as SKBR3 and BT474. The evaluation was conducted in comparison to the non-vectorized nanoparticles (MZ1-NPs) and particularly to free MZ1 treatments (Figure 3). This effect is especially relevant in the most resistant cell line, SKBR3, where ACNPs achieved a reduction in cell viability over 50% in a dose-dependent manner (Figure 3A). This promising effect at low doses, together with the fact that trastuzumab will guide the nanoparticle to the HER2 expressing cancer cells, makes this approach promising. Remarkably, no additional toxicity of MZ1-ACNPs was observed when compared to free MZ1, suggesting that the ACNPs are not toxic and cannot exert any additional or side effects when used as vehicles for the drug.

In terms of their mechanism of action, MZ1-NPs and MZ1-ACNPs did not show a strong effect in the cell cycle distribution, and their effect followed the same pattern than the free MZ1 (Figure 4A,B). The effect of MZ1 in a TNBC context was previously reported by our laboratory, where free MZ1 was not able to modify cell cycle phases [21] significatively. Of note, conjugation did not modify the drug mechanism of action. In contrast, the increment in toxicity was correlated by the induction of apoptosis (Figure 4C,D).

MZ1-ACNPs also rendered a strong cytotoxic effect against MZ1 primary resistant cell line, HCC1954 (Figure 5). Conjugation strategy could considerably improve the capability of internalization, therefore, improving the drug release in the site of action, which in turn could justify the strongest cytotoxic effect observed when compared to the same concentration of free MZ1. However, further studies should be performed to evaluate the mechanism of cell penetration. These findings offer new therapeutic opportunities for PROTACs. In clinic, these novel ACNPs could be suitable to be used in patients whose tumors were MZ1 intrinsically or de novo resistant. Although MZ1 is proposed as a therapeutic option for HER2+ breast cancer, other HER2+ solid tumors should be explored in vivo to position these ACNPs as versatile nanomedicines in oncology.

## 5. Conclusions

The use of ACNPs for the delivery of PROTACs may reduce time-consuming investigations regarding the structure–activity relationships. Despite the lack of in vivo data described in this work, we deem the ACNPs would improve pharmacokinetics parameters such a longer plasma half-life and slower elimination rate in circulation. On the other hand, the low cargo of the MZ1 into the ACNPs, the high cost of their manufacturing are challenges to investigate for clinical translation of MZ1-ACNPs. Further investigation should be pursued.

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
