# Peer review of "Controlled Delivery of BET-PROTACs: In Vitro Evaluation of MZ1-Loaded Polymeric Antibody Conjugated Nanoparticles in Breast Cancer"

_pharmaceutics, 2020, doi:10.3390/pharmaceutics12100986_

Round 1

Reviewer 1 Report

In the manuscript entitled ‘Controlled Delivery of BET-PROTACs: in vitro evaluation of MZ1-loaded polymeric antibody conjugated nanoparticles in breast cancer’ the Authors report the synthesis and application of antibody conjugated MZ1 loaded nanoparticles based on two polymers PLA and PEI. These nanoparticles were characterized by means of size, polydispersity, and zeta-potential. The morphology of the nanoparticles was analyzed by TEM. They tested prepared nanoparticles for their cytotoxic effect and its mechanism of action.
The topic is interesting since polymeric nanoparticles are a promising platform for nanomedicine, but I found several weaknesses that have to be addressed before publication.

1. The manuscript needs the scheme to show the concept of particle design and synthesis methods.
2. Please provide the size distribution curve for tested nanoparticles
3. Results of characterization of the synthesized nanoparticles have to be checked and verified, e.g. why on TEM images size of presented NPs are below 200nm while the information provided in Table 1 indicates that the diameter of synthesized NPs is bigger than 200nm,
4. Why antibody conjugated NPs are smaller than NPs without antibody
5. TEM images of both types of NPs should be presented
6. PDI reported in Table 1 is very low indicates monodisperse systems, while on TEM picture polydisperse system is presented, moreover on the TEM picture aggregated NPs are presented
7. In the Method section, authors wrote that stability was tested in the presence of 10% of serum, while in the Results section they presented results for only PBS
8. Figure 2 what is A and what is B?
9. There are not enough presented results supported the conjugation of antibody to NPs' surface, zeta potential can be influenced by many factors e.g. type of ions, ionic strength, etc…
10. Why release of MZ1 from antibody conjugated NPs is slower?
11. Why there is no dose-dependent activity of MZ1 loaded ACNPs in case of BT474 cell line

Author Response

Reviewer 1. In the manuscript entitled ‘Controlled Delivery of BET-PROTACs: in vitro evaluation of MZ1-loaded polymeric antibody conjugated nanoparticles in breast cancer’ the Authors report the synthesis and application of antibody conjugated MZ1 loaded nanoparticles based on two polymers PLA and PEI. These nanoparticles were characterized by means of size, polydispersity, and zeta-potential. The morphology of the nanoparticles was analyzed by TEM. They tested prepared nanoparticles for their cytotoxic effect and its mechanism of action.
The topic is interesting since polymeric nanoparticles are a promising platform for nanomedicine, but I found several weaknesses that have to be addressed before publication.

1. The manuscript needs the scheme to show the concept of particle design and synthesis methods.

Response. Scheme 1 in the revised manuscript depicts the synthetic methodology of the nanoparticles.

2. Please provide the size distribution curve for tested nanoparticles

Response. Size distribution curves for MZ1-NPs and MZ1-ACNPs are shown in Figure S1 and S2 of the Supporting Information, respectively.

3. Results of characterization of the synthesized nanoparticles have to be checked and verified, e.g. why on TEM images size of presented NPs are below 200nm while the information provided in Table 1 indicates that the diameter of synthesized NPs is bigger than 200nm,

Response. Table 1 collected the average size (diameter, nm), as it was depicted in the table heading of the first version of the manuscript. However, there was a mistake in the title table which indicated hydrodynamic radio. It has been corrected in the revised manuscript. In addition, another TEM image of the entire nanoparticle replaces the one in the first version.

The same mistake is corrected in figure 2B. Average size (diameter in nm) replaces hydrodynamic radio (RH in nm)

4. Why antibody conjugated NPs are smaller than NPs without antibody.

Response. The conjugation strategy requires the use of the NPs suspension in PBS (pH 5.8). Such acid media might slightly erode the surface of the NPs previous conjugation, which could explain the smaller hydrodynamic radio of the MZ1-ACNPs in comparison to MZ-NPs.

5. TEM images of both types of NPs should be presented

Response. Supporting information collects TEM images of MZ1-NPs (Figure S3).

6. PDI reported in Table 1 is very low indicates monodisperse systems, while on TEM picture polydisperse system is presented, moreover on the TEM picture aggregated NPs are presented

Response. Authors consider the measurements obtained by DLS more accurate to report the average size and PDI of the NPs since TEM images are obtained from different fractions of the whole suspension. On the other hand, TEM images did not show aggregates. We need to fix the NPs in a way to not damage their structure. The sample is embedded in a liquid resin such as epoxy resin. Such resin can maintain the NPs somehow together which could give rise to some confusion.

7. In the Method section, authors wrote that stability was tested in the presence of 10% of serum, while in the Results section they presented results for only PBS

Response. It was just a mistake. The experimental section has been corrected in the revised manuscript.

8. Figure 2 what is A and what is B?

Response. A and B have been eliminated in the revised manuscript. Both images were ACNPs samples.

9. There are not enough presented results supported the conjugation of antibody to NPs' surface, zeta potential can be influenced by many factors e.g. type of ions, ionic strength, etc…

Response. Standard protocol of Bradford assay was employed for quantifying the concentration of the antibody in the supernatant. 1.6 nM was the trastuzumab cargo over the NP surface selected according our previous published results (https://doi.org/10.3390/nano9121793). A sentence is added in the revised manuscript to avoid confusion and procedure reported in the materials and methods section.

10. Why release of MZ1 from antibody conjugated NPs is slower?

Response. The antibody conjugation over the surface of the nanoparticles might delay the diffusion of the drugs due to the most difficult accessibility of the water molecules to the core of the nanoparticles. A short paragraph is added in the results section of our revised manuscript.

11. Why there is no dose-dependent activity of MZ1 loaded ACNPs in case of BT474 cell line

Response. As it can be seen in Figure 3, encapsulated and free MZ1 exerted dose-dependent effect over the viability in both cell line. Given the fact that BT474 are more sensitive than SKBR3 and MZ1 displays very high cytotoxicity, no additional effect by increasing the drug concentration were observed for BT474.  

Reviewer 2 Report

The manuscript would be of interest to the audience of journal. It’s recommended that this contribution be accepted after major revision.

The following minor revision is suggested before acceptance for publication:

- What’s the drug release kinetics under acid pH?

- Experiments of protein corona analysis should be added.

- The authors should do experiments of NPs cellular quantification.

Author Response

Reviewer 2. The manuscript would be of interest to the audience of journal. It’s recommended that this contribution be accepted after major revision.

The following minor revision is suggested before acceptance for publication:

- What’s the drug release kinetics under acid pH?

Response. The release kinetic under pH 5.8 is added to the Supporting Information of our manuscript. In addition, it is indicated in the results section.

- Experiments of protein corona analysis should be added.

Response. Protein corona analysis was carried out. The results are depicted in the results section and the procedure in the materials and methods section. Besides, Figure S5 illustrates the results of the experiment.

- The authors should do experiments of NPs cellular quantification.

Response. Because neither nanoparticles nor MZ1 were fluorescent, the authors did not consider NPs cellular quantification. A methodology which does not require the use of fluorescent nanoparticles were followed and results are shown in the results section of the revised manuscript. In the same way, the materials and methods section explains the methodology followed.

Reviewer 3 Report

Controlled Delivery of BET-PROTACs: In Vitro Evaluation of MZ1-loaded Polymeric Antibody Conjugated Nanoparticles in Breast Cancer is a very interesting research work demonstrating the potential of the use of trastuzumab-conjugated nanoparticles prepared from PLA and PEI to deliver MZ1 PROTAC for the treatment of breast cancer. This research work shows really promising results, mainly taking into account that the proposed treatment could be applied to HER2+ breast cancer that is still considered as incurable. However, some improvements are suggested to authors:
1. Could you please further explain why TKIs generates drug resistance while
PROTACs do not?
2. Line 59: the reference 21 is in uppercase
3. Line 71: ACPNs acronym was already described in the abstract
4. Line 84: Poly-D-L-lactide
5. Lines 96 and 97: you say that the “NMR spectra were recorder on a”. Could you further explain this sentence, please?
6. Line 97: I would suggest to delete the sentence “The morphology of the nanoparticles was characterized by using transmission electron microscope (TEM)” since it is repeated in the next lines
7. Why do you use low temperature when centrifuging loaded NPs?
8. Line 106: I guess it should say “The acetone from” instead of “The acetone form”
9. Line 119: I guess it you say “10 mL of” instead of “10 mL de”
10. In the Materials section, you say you prepare your own PLA, however there is no confirmation of its characterization, mainly the polymer’s Mw or Mn. I would suggest including polymer characterization at least in a Supplementary Section.
11. How could you explain that the hydrodynamic radio of the NPs is higher than the one showed by the ACNPs?
12. Could you please further explain the methodology followed to obtain the EE and LE? A discussion about the appropriateness and validity of these results would be appreciated
13. Line 261: I guess it should say “the low values” instead of “The slow values”
14. Would you have already in mind any way to increase the ACNPs’ cargo?

Author Response

Reviewer 3. Controlled Delivery of BET-PROTACs: In Vitro Evaluation of MZ1-loaded Polymeric Antibody Conjugated Nanoparticles in Breast Cancer is a very interesting research work demonstrating the potential of the use of trastuzumab-conjugated nanoparticles prepared from PLA and PEI to deliver MZ1 PROTAC for the treatment of breast cancer. This research work shows really promising results, mainly taking into account that the proposed treatment could be applied to HER2+ breast cancer that is still considered as incurable. However, some improvements are suggested to authors:

  1. Could you please further explain why TKIs generates drug resistance while
    PROTACs do not?

Response. A classical mechanism of resistance to kinase inhibitors is the presence of primary or secondary mutations in the kinase domain that decrease or prevent the binding of the compound in the ATP pocket. For instance, mutations in Brutons tyrosine kinase are involved in resistance to ibrutinib, an inhibitor of this kinase that is approved for the treatment of several haematological malignancies such as relapse/refractory mantle cell lymphoma, chronic lymphocytic leukemia and Waldenström macroglobulinemia (Honigberg, L.A. et al. (2010) The Bruton tyrosine kinase inhibitor PCI-32765 blocks B-cell activation and is efficacious in models of autoimmune disease and B-cell malignancy. Proc. Natl. Acad. Sci. U. S. A. 107, 13075–13080. Woyach, J.A. et al. (2014) Resistance mechanisms for the Bruton’s tyrosine kinase inhibitor ibrutinib. N. Engl. J. Med. 370, 2286–2294). Analogously, it is well known that mutations in the chimeric oncogene BCR/ABL cause resistance to tyrosine kinase inhibitors used in chronic myeloid leukemia (Pagliarini, R. et al. (2015) Oncogene addiction: pathways of therapeutic response, resistance, and road maps toward a cure. EMBO Rep. 16, 280–296). In the case of solid tumors, mutations in the EGFR, such as the T790M have been associated to resistance to first generation EGFR kinase inhibitors such as gefitinib or erlotinib. For these diseases in which tyrosine kinases play a pathophysiological role, development of PROTACs with a warhead able to bind the mutated kinase, for example at an allosteric site, could result in a stable interaction potentially rescuing the resistance (Tinworth, C.P. et al. (2019) PROTAC-Mediated Degradation of Bruton’s Tyrosine Kinase Is Inhibited by Covalent  Binding. ACS Chem. Biol. 14, 342–347). Resistance due to mutations in proteins which lack enzymatic activity can also be bypassed by PROTACs. Thus, PROTACs targeting the ER in breast cancer could rescue resistance to anti-estrogens when this resistance is mediated by mutations at the ER, supporting the development of ER PROTACs in this situation (John J. Flanagan, Yimin Qian, Sheryl M. Gough, ARV-471, an oral estrogen receptor PROTACTM protein degrader for breast cancer. SABCS Dec 4-8 2018 P5-04–18).

A paragraph about in the introduction section is added in the revised manuscript.

  1. Line 59: the reference 21 is in uppercase

Response. It is corrected in the revised manuscript.

  1. Line 71: ACPNs acronym was already described in the abstract

Response. It is corrected in the revised manuscript.

  1. Line 84: Poly-D-L-lactide

Response. It is corrected in the revised manuscript.

  1. Lines 96 and 97: you say that the “NMR spectra were recorder on a”. Could you further explain this sentence, please?

Response. It is corrected in the revised manuscript.

  1. Line 97: I would suggest to delete the sentence “The morphology of the nanoparticles was characterized by using transmission electron microscope (TEM)” since it is repeated in the next lines

Response. It is deleted in the revised manuscript following referee suggestion.

  1. Why do you use low temperature when centrifuging loaded NPs?

Response. If the polymer nanoparticle suspension is heated up toward Tg then the polymer will behave in the rubbery or viscous state and the particles will aggregate (Tweedie, C.A., Constantinides, G., Lehman, K.E., Brill, D.J., Blackman, G.S., and Van Vliet, K.J., “Enhanced stiffness of amorphous polymer surfaces under confinement of localized contact loads,” Advanced Materials. 19 2540-2545, 2007).

  1. Line 106: I guess it should say “The acetone from” instead of “The acetone form”

Response. It is corrected in the revised manuscript.

  1. Line 119: I guess it you say “10 mL of” instead of “10 mL de”

Response. It is corrected in the revised manuscript.

  1. In the Materials section, you say you prepare your own PLA, however there is no confirmation of its characterization, mainly the polymer’s Mw or Mn. I would suggest including polymer characterization at least in a Supplementary Section.

Response. 1H and 13C NMR of the PLA samples are included in the Supporting Information.

  1. How could you explain that the hydrodynamic radio of the NPs is higher than the one showed by the ACNPs?

Response. The conjugation strategy requires the use of a NPs suspension in PBS pH 5.8. Such acid media might slightly erode the surface of the NPs previous conjugation which could explain the smaller hydrodynamic radio of the MZ1-ACNPs in comparison to MZ-NPs

  1. Could you please further explain the methodology followed to obtain the EE and LE? A discussion about the appropriateness and validity of these results would be appreciated

Response. The methodology section regarding EE and LE values calculation are extended in the materials and method section. Besides, a discussion of the meaning of these values is added in the discussion section of the revised manuscript.

  1. Line 261: I guess it should say “the low values” instead of “The slow values”

Response. It is corrected in the revised manuscript following reviewer suggestion.

  1. Would you have already in mind any way to increase the ACNPs’ cargo?

Response. Authors would like to explore other strategies for conjugation. Covalent strategies such as the use of click chemistry and maleimide chemistry to get ready the polymer for conjugation bear in mind. For that purpose, the catalysis will help use for the design of the appropriate modified PLA. Other option is the use of PLA-PEG polymer to increase the hydrophilicity of the polymers. This feature could also improve ACNPs’ cargo.

Round 2

Reviewer 1 Report

The topic is interesting since polymeric nanoparticles are a promising platform for nanomedicine, but the manuscript still requires modification.
1. The presented scheme has to be redrawn, the scheme of PEI is unacceptable, it looks like surfactant or something similar while PEI is linear or branched polymer, what represents red head? The scheme proposed side on adsorption, why? A positive charge is more less equally distributed in this polymer.
2. Data presented in Figure S3 do not match to data presented in Table 1, Which NPs are bigger?
3. Both TEM images should be in the original manuscript, moreover, additional discussion of what is visible should be added. When you look at image A in SM, there are also visible ‘tabs’ at the surface of NPs, is it antibodies? Or what? NPs need additional characterization.
4. The size of NPs before conjugation to antibodies should be measured in the same condition.
7. Description of the stability test was corrected however title not ‘Stability of NPs in human serum’
9. In the reviewer opinion additional work to prove antibody immobilization, as well as their activity, should be performed.
10. The authors claimed that antibodies immobilized at NPs surface influence release due to the most difficult accessibility of the water molecules to the core of the nanoparticles, what is a surface covered by immobilized antibodies? Based on the provided TEM it is not so high.

In my opinion, additional characterization of NPs has to be performed.

Author Response

Please, attached the rebuttal letter of the second round.

Reviewer 2 Report

Thanks for reply

Author Response

Thank you for your comments.

Round 3

Reviewer 1 Report

The manuscript still requires modification. The part concerning the characterization of the nanoparticles is far from perfect. There are a lot of weak points. The most important are size measurements and evidence for antibody immobilization.

1.From the previous round of revision

Data presented in Figure S3 do not match to data presented in Table 1, Which NPs are bigger?

Response. The authors used DLS measurements to report the average size and polidisperties of the nanoparticles. TEM images were only performed to study the morphology of the formulations and observe antibody conjugation. Therefore, only images from specific places were taken. However, and following the reviewer suggestion, we have included other TEM images for MZ1-NPs which could match better with the average size reported in Table 1.

The question concerned size distributions of MZ1-NPs and MZ1-ACNPs presented in figure S3 and values presented in Table 1 (both from DLS measurements). It had nothing to do with TEM images.

  1. Both TEM images should be in the original manuscript, moreover, additional discussion of what is visible should be added. When you look at image A in SM, there are also visible ‘tabs’ at the surface of NPs, is it antibodies? Or what? NPs need additional characterization.

 Response. As it was explained above, other TEM images replace those showed in the first round to help the discussion. There are not tabs in the NPs in the case of MZ1-NPs, as it can be clearly observed when compare TEM images of both formulations in the second round of the revised manuscript. Images of MZ1-NPs were moved from the supporting information to the original manuscript. Antibody conjugation was supported by BCA analyses, TEM images and results from MTTs assays. At this moment in our lab, we have conjugated fluorescent trastuzumab to the nanoparticles to study the uptake of the formulations. Even though we are still working on the optimization of the methodology, from some preliminary results the reviewer could observe the fluorescence of the nanoparticles. Neither the polymer or MZ1 are fluorescent. Therefore, the fluorescence observed is due to the conjugation of the trastuzumab over the surface of the nanoparticles. Unfortunately, we cannot add this information in our revised manuscript. They are only preliminary results.

It is not allowed to change TEM images only in order to support your statement, using other words, to have choosen only those that match your vision. Which one present real samples? Moreover, BCA and MTT are not direct and sufficient methods of analysis of antibodies immobilization. Since fluorescently labeled trastuzumab is available in your lab there is a simple way to confirm immobilization by fluorescent measurements,  it should be performed to support the immobilization process and it should be included in this manuscript. Without proper analysis, you can only suggest immobilization.

  1. Question from the first report

Why antibody conjugated NPs are smaller than NPs without antibody.

Response. The conjugation strategy requires the use of the NPs suspension in PBS (pH 5.8). Such acid media might slightly erode the surface of the NPs previous conjugation, which could explain the smaller hydrodynamic radio of the MZ1-ACNPs in comparison to MZ-NPs.

The size of NPs before conjugation to antibodies should be measured in the same condition.

Response. The authors do not understand very well the request of the reviewer. ACNPs and NPs were collected after centrifugation and the size measured by DLS. Both NPs were also lyophilised and suspended afterwards in PBS. The average size was also measured, and the data were not significant different from the other measurements.  

In the case of properly performed DLS measurement, they could also support antibody immobilization. Generally, there is chaos in presented results concerning the physical-chemical characterization of NPs.

  1. Description of the stability test was corrected however title not ‘Stability of NPs in human serum’

Response. It was corrected in the revised manuscript.

Since you tested protein corona formation, why You didn’t measure the size of NPs after incubation with proteins? Moreover, results presented in figure S5 are strange, both NPs are positive, more than +30mV. To prevent nonspecific adsorption of proteins charge of NPs has to be screened by e.g. uncharged polymers, here in both cases electrostatic interaction between NPs and proteins should be similar. Please use the same abbreviations in the manuscript as well as in SUPPORTING INFORMATION e.g. figure S5